# The evolution of dam induced river fragmentation in the United States

**Rachel A. Spinti** [1] ✉, **Laura E. Condon**[1] ✉ & **Jun Zhang**[2]

It is established that dams decrease river connectivity; however, previous global scale studies of river fragmentation focused on a small subset of the largest dams. In the United States, mid-sized dams, which are too small for global databases, account for 96% of major anthropogenic structures and 48% of reservoir storage. We conduct a national evaluation of the evolution of anthropogenic river bifurcation over time that includes more than 50,000 nationally inventoried dams. Mid-sized dams account for 73% of anthropogenically created stream fragments nationally. They also contribute disproportionately to short fragments (less than 10 km), which is particularly troubling for aquatic habitats. Here we show that dam construction has essentially reversed natural fragmentation patterns in the United States. Prior to human development, smaller river fragments and less connected networks occurred in arid basins while today we show that humid basins are the most fragmented due to human structures.

Today there are more than 50,000 medium to large dams impeding rivers across the contiguous United States (CONUS)[1]. It is well established that dams fragment river networks and alter natural streamflow dynamics, changing the flow of organisms, nutrients, sediments, and contaminants through these systems[2–5]. However, previous large-scale analyses of river fragmentation have relied on global dam databases that include only reservoirs with more than 100 million cubic meters (MCM) of storage[3,6–8]. In the US, this approach misses 96% of the nationally inventoried structures.

Watershed connectivity (i.e., the ability for water to flow freely through a system without anthropogenic intervention) is a key metric of ecosystem health. In natural systems, river fragments extend from the headwaters to the outlet or terminal point of the river. Undammed systems transport water, organisms, nutrients, sediments, and energy to sustain aquatic habitats[3,9]. Seasonal flooding in unregulated systems is also critical to the redistribution of nutrients and sediments across watersheds[10].

Dams fragment river systems by interrupting watershed connectivity. Anthropogenic river fragmentation is the increase in the number of river fragments and changes in river fragment length that are directly caused by constructed flow barriers. Because dams inherently obstruct and alter natural streamflow dynamics, they disrupt connectivity and create river fragmentation[3,5–8,11–14]. River fragmentation caused by anthropogenic structures threatens freshwater biodiversity[15,16], and has been linked to a loss of freshwater fish populations[8] (particularly salmonoid species[16]). The negative impacts of dams on ecosystems can be felt both upstream and downstream of the structures because of linkages between headwater systems and lower reaches[6,9,16,17].

We know that river fragmentation is a widespread problem. Global studies estimate that 63% of the world's longest rivers are no longer free flowing, and 48% of all river reaches worldwide have some level of diminished river connectivity[3]. Impacts are even greater in the US, which stands out as a heavily regulated system in global studies[3,6,8,11].

Furthermore, authors of previous global studies acknowledge that their results are conservative estimates because they focus only on large dams that are included in global datasets (i.e. dams with more than 100 MCM of storage)[3,6–8]. While it is true that large dams account for most reservoir storage and play a large role in the regulation of major river systems, the vast majority of structures are smaller and their cumulative impact on the river network can be significant[4,18]. In the US, there are 1945 large dams in the global database, which is a small fraction of the more than 50,000 major structures documented in the National Anthropogenic Barrier Dataset (NABD)[1].

[1]Hydrology and Atmospheric Sciences, University of Arizona, 1133 James E. Rogers Way, Tucson 85721 AZ, USA. [2]Key Laboratory of VGE of Ministry of Education, Nanjing Normal University, Nanjing 210000, China. ✉e-mail: rspinti@arizona.edu; lecondon@arizona.edu

Here we show how river fragmentation in the US has evolved over time as a result of mid-sized and large dam construction. We explore trends in the total regulatory capacity on rivers, the number of fragments, and the fragment length caused by these structures. We also explicitly separate mid-sized dams from the large dams used in global studies to quantify the extent to which global approaches under-represent human impacts.

## Results

### Mid-sized dams greatly outnumber large dams

Here we evaluate the fragmentation caused by 51,923 medium to large structures in CONUS over time. Our analysis is based on the National Anthropogenic Barrier Dataset (NABD)[1], which is a mapping product derived from the National Inventory of Dams (NID)[19]. Previous studies have used NABD to investigate the impacts of dam removal[20], calculate dam metrics, and explore the effect of large dams on fish habitats and population[21,22]. Cooper et al.[21] used the NABD database to calculate 21 different dam impact metrics across the CONUS including river fragmentation and degree of regulation in their analysis of dam impacts on fish assemblages. Our approach is methodologically very similar to theirs (both with respect to metrics and underlaying datasets), but considers evolving impacts over time.

The NID includes all structures that pose high downstream flooding risk if the dam fails or that have a low risk but meet a minimum size requirement (at least 7.6 m high and more than 18,500 cubic meters of storage or more than 1.8m high with storage exceeding 61,700 cubic meters). Dam size definitions vary, but for the purposes of our analysis, we define all structures meeting the NID criteria but storing less than 100 MCM as medium- (or mid-) sized dams.

The relative importance of medium dams varies by river basin, but in all major river basins (Fig. 1a), medium dams (Fig. 1b) account for at least 80% of the nationally inventoried structures (Fig. 1c). The importance of medium dams is generally greater in the more humid regions of the country. In our largest river basin (the Mississippi), mid-sized dams account for 97% of structures (Fig. 1c). Mid-sized dams also account for 48% of storage nationally and in some basins, such as the Rio Grande and Columbia, as much as 80% and 50% respectively (Fig. 1d).

Our analysis expands significantly on the dams included in previous global fragmentation studies; however, it is still limited to nationally inventoried dams meeting the NID threshold criteria. Graf (1993)[23] estimates there may have been as many as 2 million dams built in the US at one point, most of which are small structures. Unfortunately, information on these smaller structures is generally maintained by state regulatory agencies, and data coverage and accessibility are uneven or non-existent. Poff & Hart (2002)[4] compared the coverage of the NID to state documentation for Wisconsin and Utah and found that the national database included 17% and 6% of the total dams inventoried by the states respectively. Lacking reliable mapping and size reporting of small structures, we are unable to include them in quantitative fragment analysis.

### Regulation has steadily increased over time

Degree of Regulation (DOR), defined as the ratio between upstream storage capacity and annual stream flow volume, has steadily increased over time. Before 1920 (Fig. 2a), most river networks had no dam regulation, and for those with dams, the DOR was less than 0.2. Only one tributary of the Missouri was an outlier with a DOR greater than 0.6. Prior to 1920, dam construction primarily occurred along the main stems of rivers, particularly the Missouri, Mississippi, Columbia, Rio Grande, and some of their tributaries (Fig. 2a). Between 1920 and 1950, regulation increased along the main stems of the major river networks and some of their major tributaries (Fig. 2b). By 1950, there were major reaches where storage exceeded annual flow (i.e., DOR > 1). By 1980, the DOR of most major river networks was greater than 0.4 and regulation had expanded into the headwaters of those networks and their tributaries (Fig. 2c). As of 2010, large portions of the Mississippi and Colorado basins had DOR above 1, and almost no mainstem reaches had DOR less than 0.1 (Fig. 2d). Of course, there were some dam removals that occurred over this time period, but the overriding trend was toward increased development.

It should also be noted that DOR is a metric indicating the potential for a dam to regulate a streamflow regime based purely on the dam's storage volume. Actual impacts on the streamflow regime will depend heavily on operating policies and reservoir purpose[4].

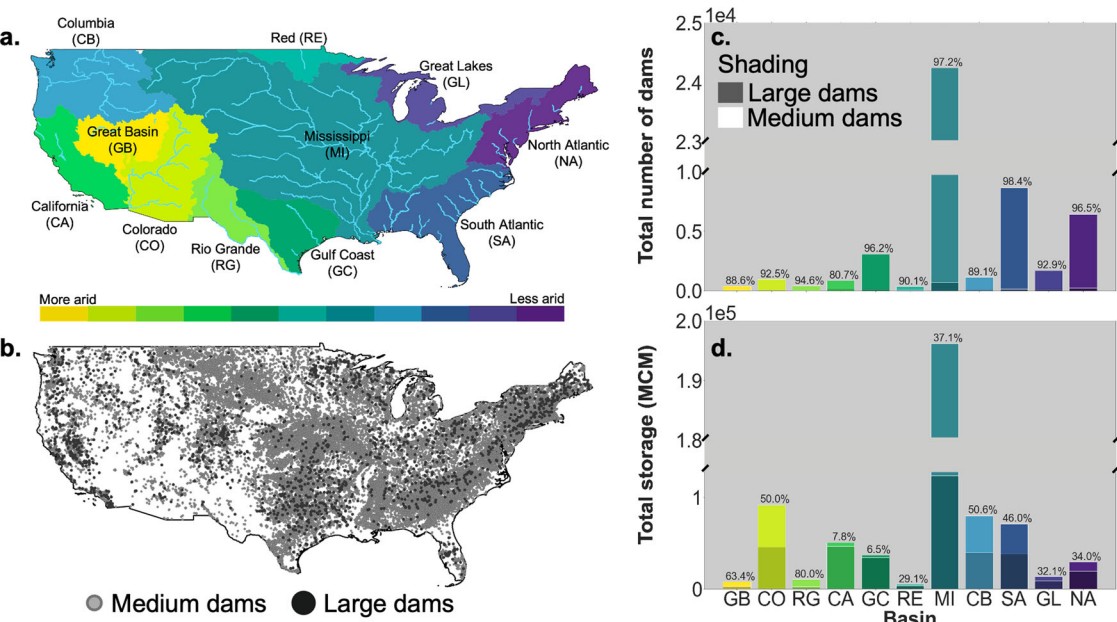

**Fig. 1 | Comparison of number and reservoir storage capacity of medium vs. large dams. a** Map of the contiguous United States (CONUS) with major rivers and major river basin aridity. **b** Map of CONUS with large dams and medium dams. **c** The total number of dams present in each major river basin. **d** Total reservoir storage present in each major river basin. The percentage of medium dams contributing to each total are placed on top of the bars in **c** and **d**.

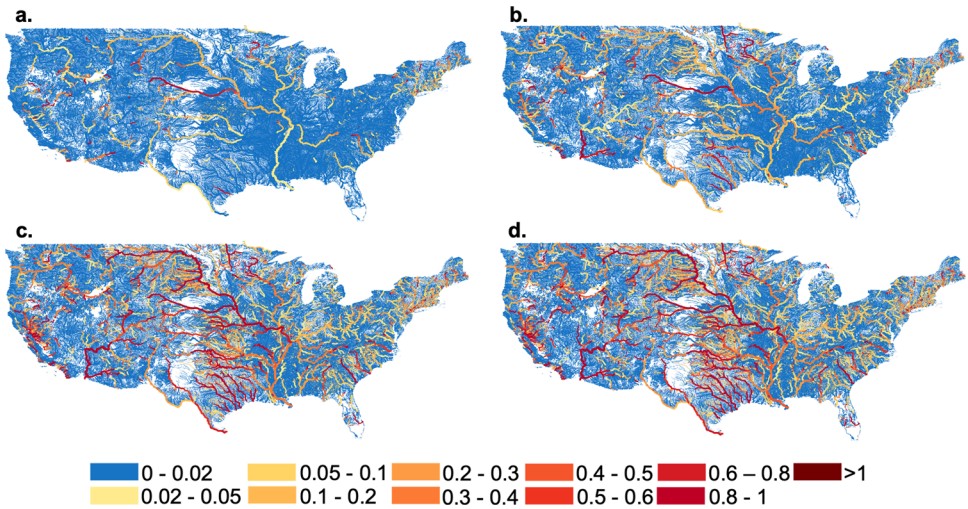

**Fig. 2 | Expansion of dam-based river regulation into tributaries and headwater systems over time.** Maps of the Degree of Regulation (DOR) (storage capacity divided by total annual streamflow) over time calculated on the National Hydrography Dataset Plus Version 2 stream network over time[27]. **a** 1920, **b** 1950, **c** 1980, **d** 2010. As DOR increases, the stream network transitions from blue to yellow to red. Stream networks with DOR greater than one are shown in dark red. The line width varies based on Strahler stream order[30] such that river segments with larger stream orders have thicker lines.

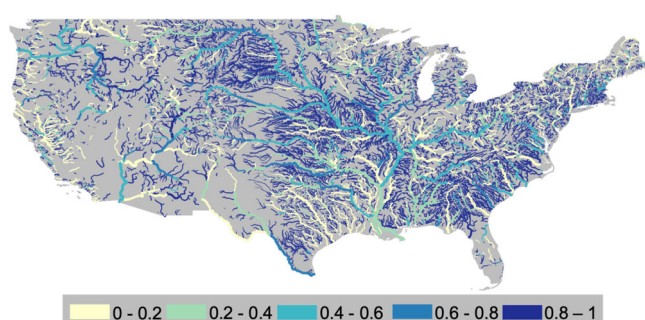

**Fig. 3 | Medium dams regulate headwater systems.** The fraction of DOR caused by medium dams as of 2010.

Furthermore, regulatory impacts of a dam evolve over time if operating policies change.

## Medium dams have more regulatory potential than large dams

Medium dams account for at least 10% of total DOR in every region of the US, but in some places, they contribute more than 80% (Fig. 3). The DOR contributed by medium dams decreases from east to west. Along the East Coast, at least 40% of the river regulation comes from medium dams. This trend extends into parts of the Midwest and Plains, especially along the Missouri River. The impact of medium dams decreases moving to the lower sections of the Mississippi because their storage is compensated for by increasing streamflow moving downstream. In the western US, the contribution to DOR from medium dams is generally below 40%. In the west, large dams are prevalent and medium dams are often located further downstream because it takes a larger drainage area to support a reservoir in a more arid climate.

Medium dams contribute the most to the DOR of headwater systems and tributaries. For most major river systems (e.g., Colorado, Columbia, and Missouri), medium dams account for more than 80% of headwater regulation (Fig. 3). Along major rivers, the impact from mid-sized dams generally decreases moving downstream. For example, along the Columbia, the contribution from medium dams decreases from at least 80% in the headwaters to roughly 40% at the outlet.

However, there are exceptions to this pattern especially in the Southwest. Along the main stems of the lower Colorado and Rio Grande, the impact of medium dams increases moving downstream. This pattern is caused by dams located on tributaries that flow into downstream portions of the river. In more arid environments, medium dams can operate in larger drainage areas than in humid environments because the annual flows are smaller.

## Dams reshaped aridity and fragment density patterns

Prior to human development, there was natural diversity in river fragmentation and fragment lengths caused by aridity and topography. Natural fragmentation was largest in arid regions, particularly in the Great Basin, the Lower Colorado, the Rio Grande, and along coastlines (Fig. 4a). The natural fragmentation in arid basins is due to ephemeral river reaches and natural sinks where there is not enough annual precipitation to support perennial flows on a fully connected drainage network. The coastlines have high fragment densities because flowlines end in the ocean or the Great Lakes. Outside of these areas, the remainder of the US was largely covered by fully connected drainage networks with little or no fragmentation prior to development.

Dams have reshaped these patterns. By 1920, the impact of human development on river connectivity is already clear (Fig. 4b). The East Coast became heavily fragmented as did the lower Mississippi and parts of the High Plains. Fragmentation increased over time, especially in the period from 1950 to 1980 (Fig. 4c, d). By the 1980s, almost every watershed in the US had some level of fragmentation above natural conditions. Today (Fig. 4e), the spatial diversity of pre-development fragmentation is completely gone, replaced by larger fragment densities in nearly every watershed.

Fragment density changed more in humid basins, with less noticeable changes in the most arid basins (Fig. 1a). The most humid basins (e.g., the North and South Atlantic) began with low fragment density pre-development (Fig. 4a), but are now some of the most densely fragmented (Fig. 4e). The largest change in fragment density in the North Atlantic basin occured between pre-development and 1920. The Gulf Coast and Mississippi basins experienced a large jump in fragment density during the dam-building peak of 1950 to 1980 (Fig. 4c, d). These basins had the smallest fragment densities pre-development but the fourth and fifth highest fragment densities in 2010.

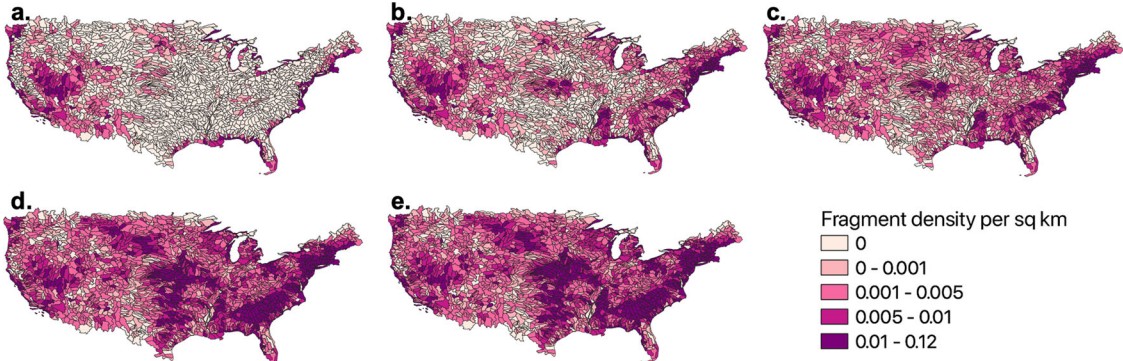

**Fig. 4 | Reduction of river connectivity over time.** Temporal change in the fragment density per Hydrologic Unit Code 8 (HUC8) basin since pre-development. **a** Number of fragments pre-development (i.e., natural fragments), **b** 1920, **c** 1950, **d** 1980, **e** 2010.

In contrast, the arid basins (Colorado, Great Basin, Rio Grande) began with moderate to high natural fragment density (Fig. 4a). Fragmentation did increase over time but less dramatically than in other parts of the country (Fig. 4a, e). Most arid basins cannot support as many reservoirs and these systems tend to be dominated by a smaller number of large reservoirs.

Nationally, we see that the spatial patterns of fragmentation have essentially been reversed by human infrastructure. In natural systems, the most arid basins are the most fragmented because they lack the water to support extensive perennial drainage networks. In heavily developed systems, humid basins become the most fragmented because there is enough water to support more structures, and there is often a need for more flood control structures. Today, there are very few watersheds remaining without some degree of added fragmentation caused by human construction. Fragmentation is no longer limited to coastal areas and arid basins but is concentrated in the Midwest and the eastern US.

### Medium dams create shorter fragments than large dams

While medium dams account for 48% of storage, they represent 96% of the structures. The relative importance of large dams for river fragmentation has declined over time. Medium dams accounted for 40% of national stream fragments in 1920, increasing to 73% by 2010 (Supplementary Fig. 2).

Figure 5 shows the change in fragmentation due to dams over time grouped by fragment length. Across all basins, the density of fragments for all length categories has generally increased over time (Fig. 5a, c, e, and g. The largest increases have occurred for fragment lengths less than 10 km (Fig. 5a), especially in more humid basins (e.g., Gulf Coast, Mississippi, North and South Atlantic), while the density of the longest fragments (greater than 1,000 km) remains more consistent between basins (Fig. 5g).

The relative importance of medium dams increases with shorter fragment lengths (Fig. 5b, d, f, h). Medium dams contribute most to fragments less than 100 km long; they account for over 70% of these fragments nationally (Fig. 5b, d). In general, short river segments often occur in headwaters or tributaries, which is also where medium dam development tends to occur. However, this relationship depends strongly on aridity. For example, in more arid basins such as California, the Great Basin, and the Rio Grande, medium dams contribute only 20% to the fragments less than 100 km in length (Fig. 5d). In the Gulf Coast and the Mississippi basins, medium dams contribute to approximately 80% of these short fragments (Fig. 5b, d).

Figure 6 illustrates that as the number of short fragments has increased, the relative frequency of long river fragments has declined. This decline is observed if we only account for large dams but is even more pronounced when we consider medium dams too (Fig. 6a–f). The largest changes in the likelihood of large fragments occur in the Gulf

Coast and Great Lakes regions (Fig. 6c, e) but shifts are observed in all basins, even in arid locations such as the Great Basin and the Rio Grande (Fig. 6a, b). This systematic shift away from long river fragments indicates a decrease in connected river habitats.

## Discussion

It is well established that anthropogenic fragmentation and regulation of river networks can significantly alter freshwater ecosystems and have contributed to the fact that freshwater species are more threatened than terrestrial species[16]. Headwater reaches support biodiversity in river networks and linkages between headwaters and downstream systems are critically important to ecosystem function[17]. While more species are threatened when dams are built on main stems[24], some researchers have argued that dams on tributaries are the most harmful for river systems as a whole because fish are unable to migrate, resulting in a decrease in biodiversity[25]. Irrespective of dam location, streamflow regulation is a major adverse ecological consequence of dams[7,12]. To fully understand the magnitude of the human impact on river networks, we need to consider all structures along the river system.

Our study of more than 50,000 medium and large dams reveals the extent of regulation and fragmentation in CONUS, which has been consistently underestimated in global fragmentation studies[3,6–8]. Analysis of mid-sized dams highlights the extent to which regulation has expanded into headwater systems over time and the compounding impact on total river regulation caused by mid-sized structures. Mid-sized dams account for 48% of total storage nationally but are the dominant source of storage in more arid basins and in headwater systems.

Medium dams are the leading driver of river fragmentation, representing 96% of the major structures in the US. In the locations with the largest change in fragment density, medium dams account for 73% of the change. It should also be noted that medium dams are often located on smaller river reaches than large dams. Thus, they contribute disproportionately to the creation of river fragments less than 100 km long.

In undeveloped systems, arid and coastal regions have the highest fragment densities. This pattern has been completely reshaped by dam construction. Today, nearly every watershed in the US has high levels of anthropogenic fragmentation, and highest fragment densities occur in humid regions. Some of this can be explained by reservoir purpose, including flood control, hydroelectricity, and water supply. The more humid eastern US has greater river density and many population centers, so there is more need for flood control. In contrast, the western US is more arid and relies on reservoirs for water supply.

This national evaluation accounts for all nationally inventoried structures that can be spatially linked to the river network. While

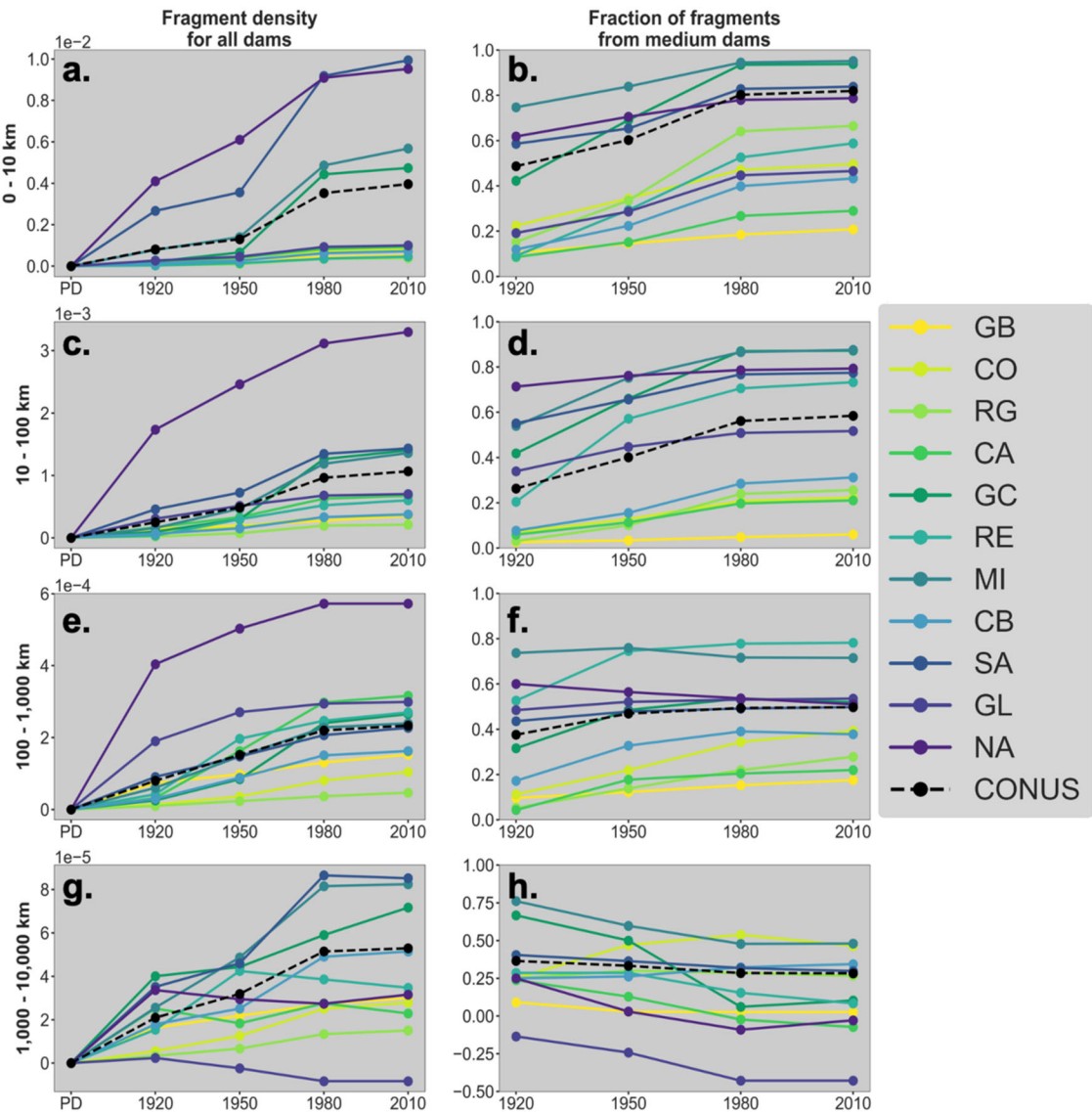

**Fig. 5 | Comparison of overall fragment density and of medium dam frequency.** Fragment density as a function of fragment length caused by all dams over time where PD is pre-development (**a**, **c**, **e**, **g**). Fraction of total fragments caused by medium dams over time (1-number of fragments from large dams/number of fragments from all dams) (**b**, **d**, **f**, **h**). Data are grouped by major river basin. The scale changes with fragment length.

NABD is more complete than other global dam databases, it is focused on dams with the largest potential to cause harm and has a lower threshold for volume and storage height. Thus, this is not a comprehensive database of all anthropogenic structures. There are likely millions of small and very small structures that further fragment our river systems[23]. Thus, our results should be viewed as a conservative estimate of river fragmentation caused by major structures.

Furthermore, since we lack operations data, we use reservoir storage volumes to calculate DOR. We use this metric as an indication of potential river regulation. However, we acknowledge that dam design, reservoir purpose, and passage systems can greatly impact the influence a structure has on a river[4]. For example, some dams have fish ladders and other structures in place to allow flow of organisms and nutrients. Similarly, the operating strategy of a flood control dam is different than that of a water supply dam. To fully understand the regulatory impact of structures on our streamflow regimes, we need a more expansive national structure inventory and accessible dam operations data.

## Methods

### Definitions

River fragmentation and regulation are two ways that dams affect our river networks. A river fragment is the portion of a river network through which water can flow freely without any anthropogenic barriers[18]. In natural systems, river fragments extend from the headwaters (which may include multiple locations) to the outlet or terminal point of the river and will likely encompass many river segments. In systems with anthropogenic barriers, a fragment may start from natural headwaters or from a dam and extend to a natural termination point or to another structure. For the purposes of this work, we define natural fragments to be fragments that end in a natural terminal point like a lake or ocean and have no barriers between outlet and headwaters. Anthropogenic fragments occur when an anthropogenic structure on a river network is constructed, becoming a barrier.

We define fragmentation to be the general pattern of river connectivity in a river basin such that increased fragmentation corresponds to decreased connectivity. Here we use the distribution and size of river fragments within a watershed to quantify the extent of

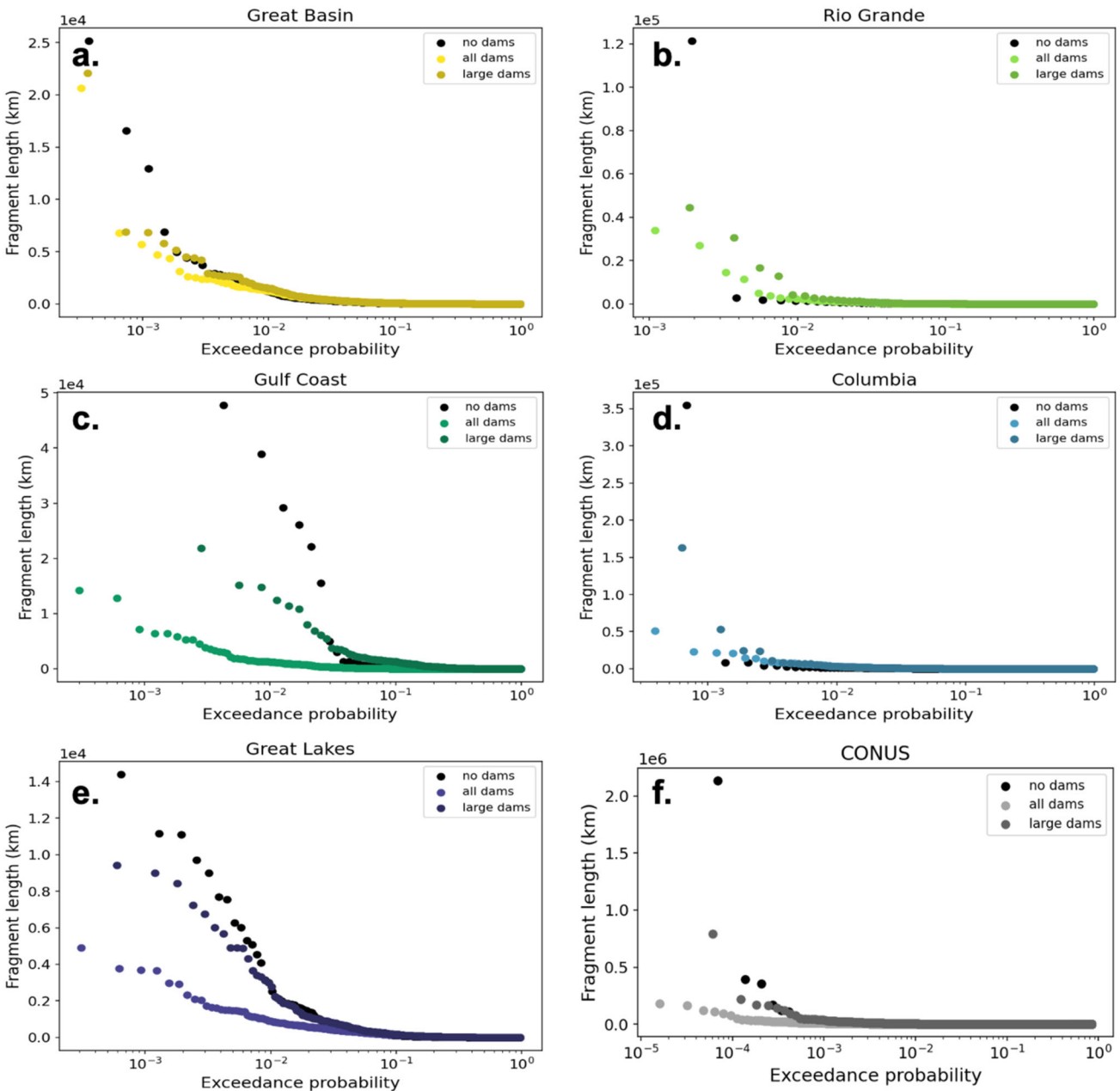

**Fig. 6 | Exceedance probability of various fragment lengths for no dam, all dam, and large dam cases in various river basins. a** Contiguous United States (CONUS), **b** Great Basin, **c** Rio Grande, **d** Gulf Coast, **e** Columbia, **f** Great Lakes.

fragmentation. Regulation comes with anthropogenic fragmentation. For our purposes, regulation is the capacity of anthropogenic structures to alter the natural flow regime. Unfortunately, actual reservoir operations are not available consistently and are especially difficult to assemble for medium structures. Therefore, consistent with previous publications, we use the degree of regulation (DOR) metric to quantify regulation. DOR is defined as the ratio of reservoir storage capacity relative to the average annual flow[7].

## Datasets
The main datasets we used in this research were the dam dataset from the National Anthropogenic Barrier Dataset (NABD) and flowlines from the National Hydrography Dataset Plus Version 2 (NHDPlusV2). We selected NABD and NHDPlusV2 for this research because they are spatially linked[1]. NABD is a revised version of the National Inventory of Dams (NID)[1], which is a census of the currently-operating dams in the

US[26]. We processed NABD in Python to remove duplicate entries and to update dams with incorrect NID identifying numbers (NIDIDs). We compared large dams from GRanD[7] and NABD to check that all large dams were included in our analysis. We imported any missing large dams from GRanD into NABD. NABD contains data such as the year that dam construction was completed (Year_comp), normal reservoir storage (Norm_stor), unique common identifier (COMID), and NIDID, all of which were used in this analysis. We created a unique dam ID (DamID) for every dam in our analysis and used it to label all fragments created by structures. Additionally, dams that are included in GRanD were flagged (Grand_flag). Roughly ~6300 dams (roughly 12% of the database and 1.2% of total storage) used in our analysis are missing year of dam construction. For our analysis we assign these dams a construction date of prior to 1920. This assumption can impact our results in some locations as dams with missing construction dates are not evenly distributed. This has a larger impact on fragmentation analysis

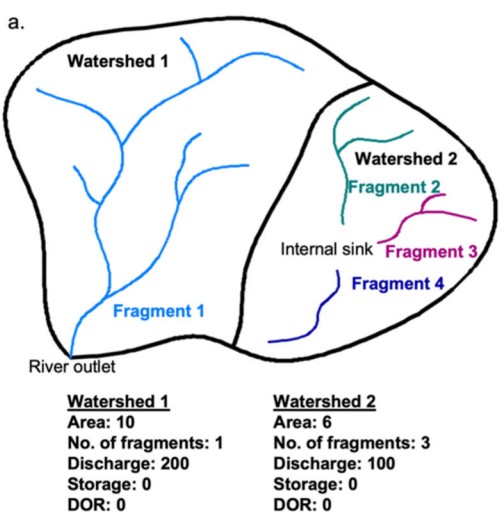

**Fig. 7 | Example watersheds to represent fragmentation algorithm.** A visual representation of how our algorithm identifies and labels fragments adapted from Grill et al.[12]. The basin has two watersheds, which are shown (**a**) pre- and (**b**) post-development. Watershed 1 contains a river network with an outlet and only anthropogenic fragmentation effects. Watershed 2 contains three river networks that drain into a sink, resulting in natural and anthropogenic effects. Fragment density and DOR were also calculated for the watersheds.

than degree of regulation as the relative storage fraction of these dams is very small.

NHDPlusV2 contains the flowlines used to indicate river reaches and tributaries[27]. It is a geospatial surface water dataset created by the US Geological Survey (USGS) in partnership with the US Environmental Protection Agency (EPA)[27]. NHDPlusV2 contains over 2.6 million flowlines, covering almost all streams in CONUS. NHDPlusV2 has a variety of attributes, which include: the length of the segment (LENGTHKM); the unique segment IDs at the current, upstream, and downstream segments (Hydroseq, UpHydroseq, and DnHydroseq); headwater indicator flag (StartFlag); type of flowline (FTYPE); unique common identifier (COMID); estimate of "natural" mean flow (QC_MA); stream order of the segment (StreamOrder); and hydrologic unit code (HUC) values. We used StartFlag to identify headwater segments and the Hydroseq attributes to determine which segments were downstream from other segments so that the river network could be traversed. We filtered out NHD flowlines that were designated as coastlines because they flowed into each other.

## Fragment processing

The purpose of fragment processing is to determine the length and location of anthropogenic and natural fragments in CONUS. Prior to fragment processing, we filtered dams by the year completed, so only the dams built prior to the specified time (pre-development, 1920, 1950, and 1980) were included. Then, we spatially joined 51,923 dams from NABD to 2.6 million NHDPlusV2 flowlines using COMID[27]. We then filtered the combined dam and flowline csv files by major river basin category (Fig. 1a) using HUC2s.

Following this initial processing, we identified and labeled anthropogenic and natural fragments. Our fragment algorithm is implemented in Python. First, it identifies all headwater segments in the watershed. Next, it loops over all headwater segments and traverses downstream from each of those segments until it reaches either (1) another river fragment that has already been processed, (2) an anthropogenic structure, or (3) the terminal point of a river. For (1), the fragment is assigned the same fragment ID as the fragment it terminates in. For (2), the fragment is assigned a fragment ID based on the dam that is its terminal point. In this case, the segment downstream of the dam is added to the list of headwater segments to be traversed. For (3), the fragment is assigned a unique identifier that starts at 999,000 by default and increases by 1 for each new natural fragment. The

algorithm works through the river network until all headwater segments have been processed. At this point, every segment in the basin has been traversed and all fragments have been assigned a unique identifier. Our fragment algorithm is designed to be consistent with the way that Grill et al.[12] define fragments in their global analysis.

Figure 7 provides an example of how the algorithm is implemented within two neighboring watersheds. Figure 7 shows Watershed 1 and Watershed 2 in their natural states (i.e., before any dam construction). Watershed 1 is a fully connected river network with a single outlet, so it comprises one fragment. Watershed 2 contains three different fragments that each end in an internal sink. Post-development, five dams have been constructed within the two watersheds (Fig. 7b). Watershed 1 now has five fragments, while Watershed 2 has four.

We calculated fragmentation metrics based on the fragments generated from the algorithm described above. The fragment density by major river basin and HUC8 was calculated by dividing the number of fragments within the basin by the area of the basin. The relative change in the number of fragments from medium dams was found by repeating our network analysis including only large dams. From this, we calculated the number of fragments which would occur in a basin given only large dams. The relative fragmentation increase caused by medium dams is calculated with the following equation.

$$\text{fragmentation increase} = \frac{1 - \#\text{of basin fragments from large dams}}{\#\text{of basin fragments from all dams}}$$

(1)

In the example watershed above (Fig. 7), fragment density more than doubled with the addition of five dams. Assuming a total drainage area of 10 for both watersheds, fragment density was 0.4 pre-development (Fig. 7a) and increased to 0.9 post-development (Fig. 7b).

## Degree of regulation (DOR)

DOR is an estimate of how a dam or set of dams alter the natural flow regime for downstream river reaches. It is the proportion of a river's annual flow that can be withheld by a reservoir or cluster of reservoirs. We calculated it using the definition from Grill et al.[3] with the normal reservoir storage obtained from NABD and the average "natural" river flow from NHDPlusV2. Normal storage is the total storage space below normal retention level in a reservoir[26]. It was used instead of maximum storage because it is more representative of average conditions. We

estimated average "natural" river flow for NHDPlusV2 using a unit runoff method and then adjusting the estimate with stream gauge data[27].

$$DOR = \frac{normal\ storage\ upstream}{average\ river\ flow} \quad (2)$$

We calculated DOR for each flowline segment from NHDPlusV2. Following Grill et al.[3], rivers with DOR less than 2% are considered to be free flowing. We calculated the fraction of DOR from medium dams by dividing the DOR of large dams by the DOR of all dams and subtracting by one for all river segments. For the example basin in Fig. 7, the addition of dams resulted in the regulation of most of the water in the basin. Pre-development DOR was 0 (Fig. 7a) because no dams were present. After five dams were constructed, DOR was 0.75 (Fig. 7b), thus there is regulation of flow.

## Data availability

This analysis is based on publicly available datasets. Links for the datasets and version numbers are included in the GitHub repository containing the data analysis code. The river fragmentation data generated in this study are available in the CyVerse Data Commons database under accession code 10.25739/bjd1-6k38 (https://datacommons. cyverse.org/browse/iplant/home/shared/commons_repo/curated/ Spinti_river_fragmentation_data_2022)[28].

## Code availability

All codes for our analysis are available from GitHub through the following repository (https://github.com/rspinti/medium_dam_ fragmentation_regulation)[29].

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

## Acknowledgements

This work was supported by the U.S. Department of Energy Office of Science, Office of Advanced Scientific Computing Research, and Biological and Environmental Sciences IDEAS project, the Sustainable Systems Scientific Focus Area under award number DE-AC02-05CH11231 (L.E.C., J.Z.); and the University of Arizona Technology and Research Innovation fund (L.E.C., R.A.S.).

## Author contributions

R.A.S. gathered all data, conducted the technical analysis, and led the manuscript preparation. J.Z. provided technical oversight on the analysis and helped with figure preparation and writing. L.E.C. wrote the fragment processing algorithm and supervised the research and revisions. She is the corresponding author.

## Competing interests

The authors declare no competing interests.
