## [Peer Review File · Nature Communications]

The evolution of dam induced river fragmentation in the United StatesREVIEWER COMMENTS

Reviewer #1 (Remarks to the Author):

The authors aim at providing the first national evaluation of the impact of small dams on river fragmentation in the contiguous US. The large-scale analyses are performed by using dam information from two datasets, the NABD and the NHDPlusV2 databases. The authors analyzed the temporal evolution of dams in the US, the contribution of small dams to river regulation, the river connectivity pattern due to small dams' presence, and the river fragmentation due to small dams.

The topic is extremely relevant and promising, the paper is written in a clear way, and the analyses on river fragmentation over time (figure 5) and exceedance probability of river fragmentation (figure 8) are interesting. However, I found the novelty and results not properly communicated and lacking in clarity.

Novelty: There is no doubt that the study has a strong environmental and societal impact. However, when reading the introduction section it is not clear what is the novelty of the paper compared to Belletti et al. (2020) and what is the methodological advance proposed by the authors to assess river connectivity/fragmentation? In the first line of the abstract, the authors mentioned that "it is well established that dams decrease river connectivity", which is also one of the main results of this study (see Figure 8). Would it be expected that small reservoirs decrease river length fragmentation considering that the high number of small dams will lead to high river fragmentation, as already shown in Belletti et al. (2020)? Despite the different geographical location, dataset, and fragmentation metrics, what is the difference between this study, Belletti et al. (2020), and other studies on the topic?

It would be really interesting to have more information on the main predictors of barrier density (e.g. agricultural pressure) and to have analyses and discussions on the environment and hydrological implications of river fragmentation. What is the effect of river fragmentation in a quantitative way (e.g. reduction of flood peak, more frequent dry periods)?

Results and discussion: I found the results reported in the abstract really vague, without a quantitative indication of the influence of small reservoirs on river fragmentation, and without a clear take-home message from the study. Regarding the discussion and introduction, I would recommend the authors to improve the literature analysis as I had the impression that the authors were biased toward the same references (e.g. Belletti et al., Grill et al., and Lehner et al.) in different parts of the paper. In the discussion section, I would include a reference at the end of line 294 to support the statement on the underestimation of regulation and fragmentation on CONUS.

Is the first section of the results (the regulation of major rivers in the US) really needed in this study? This result section reads more like a literature analysis and it could be a better fit in the introduction section to provide the reader with the importance of considering dams as drivers for river fragmentation due to their massive presence in the US (as already repeated many times in the paper)

Additional comments:

- Lines 17-29 of the introduction read more like an abstract. Of course, the way to structure the introduction section can be very subjective, but I would first introduce the main problem, then report the previous studies on the topic, then include the methodological gaps, and then describe the novelty and objective(s) of the paper.
- The current version of the paper has a total of 9 figures. If possible, I would recommend the authors to reduce the number of figures (or merge them) to keep only the most important and include the others as supplementary. For example, figure 6 is barely referred to in the paper and it could go as supplementary.
- In order to ensure transparency and replicability, data and scripts used in this study have to be reported in the corresponding sections.
- Lines 311-320 should be moved in the introduction to define since the beginning of the paper the meaning of river fragmentation also for non-experts readers.
- Figure 4. Include the years in the figure and not only in the caption
- Line 210, 73%, while 70% in line 305. Try to be consistent with the percentages

Maurizio Mazzoleni

Reviewer #2 (Remarks to the Author):

In my opinion the authors have accomplished a very good study. For a particular dataset they have performed an elegant analysis of changes in flow/storage over time, and the cutting of river segments into more and smaller links, termed river fragmentation. In essence, they have done a very nice job with an existing data base, mapping tools, and by employing some widely used metrics. Lit review, writing, analysis, are all very well done. I follow this literature, and found this manuscript to be a solid contribution.

My main concern is a sense of over-selling the significance of their study, or perhaps just that they are insufficiently forthcoming about the limitations. The analysis uses an existing data base of dams maintained by the Army Corps and digitized by Ostroff et al (2013). As the authors point out, there are just over 50,000 dams in this data base, but only 1945 of the largest are included in world data bases – hence they are performing the first analysis of “small dams” in the U.S.

The key data set is from Ostroff, A., Wiefelich, D., Cooper, A., Infante, D., & Program, U. A. G. (2013). 2013, 2012 National Anthropogenic Barrier Dataset (NABD). In. U.S. Geological Survey - Aquatic GAP Program: Denver, CO. No link is given, and it is not found in google scholar (ie. No citations to it). It can be found at:

<https://www.sciencebase.gov/catalog/item/get/512cf142e4b0855fde669828>. It was derived from point dataset of the 2009 National Inventory of Dams (NID) created by the U.S. Army Corps of Engineers (USACE). Language in the description indicates that these are all in fact large (but perhaps not “very large”) dams.

Given that Poff and Hart 2002 (not cited) estimate something of the order of two million small dams in the U.S. (and other studies point to additional barriers formed by poorly designed culverts affecting headwater streams (not cited)), the authors’ claim that they are assessing small dams, and that this is a something like a complete dataset of small dams, should be far more qualified. For comparison, the Belletti paper they cite analyzes over one million dams in Europe, further indicating the very substantial undercount in this study. Based on the authors’ presentation, readers of this study might reasonably conclude that all or nearly all small dams in the U.S. are included (Line 278 “This study is the first national evaluation which takes all structures into account.”). This is wildly untrue. It is understandable that scholars whose skill set lies in hydrological and geospatial analysis can only perform their work on suitable and available datasets. But they should not mislead their readers into thinking the underlying dataset is more exhaustive than it is. Line 293 “reveals the full extent” Using this language here is not justified.

Clearly the inclusion of an additional 48,000 dams (beyond the roughly 2,000 included in global analyses) will result in greater impacts on storage and fragmentation. However, it is worth noting that river fragmentation as used here lies in the algorithm used to estimate it. This study wishes to argue that number and placement of dams (ie, what they analyze) are more important, than size, dam design and operation, etc. They offer some caveats, then make this claim on Line 273: “Ultimately, dam impacts to ecosystems depend more on number and location of barriers, not their size (Belletti et al., 2020).” I haven’t consulted this source but am skeptical of the claim, which is presented here as received knowledge. Obviously, it supports a view that the authors’ analysis is not undermined by details of dam size, design, and operation. However, they do not make the case for this claim on line 273.

The authors do include brief mention of caveats, but in my view these are inadequate. Size of dam, design and operation, and passage devices vary considerably and are likely to profoundly influence physical (sediment transport), chemical (nutrient and materials transport) and biological (movement of organisms) dam impacts. Lines 280-288 indeed acknowledge some of these concerns, but they have the feel of sweeping under the carpet some major reservations that deserve more direct discussion.

Another concern (although not terribly serious in my view) is that the findings are not very surprising. Lines 196-200: a major finding is that river fragmentation due to dam construction increased more in humid areas than in arid areas. While nicely demonstrated by their analyses and graphics, this basically says that more dams have been built on rivers draining wetter landscapes - hardly a surprise, as the authors themselves note. Line 211 It also is unsurprising 48,000 ‘small’ dams account for more river fragmentation than 2000 ‘large’ dams – there are more of them.

Minor but perhaps worth noting: This is not the first analysis of the NABD database. I have not attempted an exhaustive search, but here I offer one example. Perhaps the authors should note such works. The 50,000 dams are mapped and their distribution discussed here (Foley MM, Magilligan FJ, Torgersen CE, Major JJ, Anderson CW, Connolly PJ, et al. (2017) Landscape context and the biophysical response of rivers to dam removal in the United States. PLoS ONE 12(7): e0180107. <https://doi.org/10.1371/journal.pone.0180107>)).

Nit-picking: Line 141 fig 4 c and 4d are mentioned before 4a and 4b

Response to comments by anonymous Reviewer 1

We would like to thank the reviewer for their constructive comments and for acknowledging the promise of our work. We have addressed their comments (bolded) below.

The authors aim at providing the first national evaluation of the impact of small dams on river fragmentation in the contiguous US. The large-scale analyses are performed by using dam information from two datasets, the NABD and the NHDPlusV2 databases. The authors analyzed the temporal evolution of dams in the US, the contribution of small dams to river regulation, the river connectivity pattern due to small dams' presence, and the river fragmentation due to small dams.

The topic is extremely relevant and promising, the paper is written in a clear way, and the analyses on river fragmentation over time (figure 5) and exceedance probability of river fragmentation (figure 8) are interesting. However, I found the novelty and results not properly communicated and lacking in clarity.

We agree that the novelty of our results was not emphasized clearly enough in the original manuscript. We have addressed this concern, and all of the other points raised by the reviewer, directly in our responses below.

Novelty: There is no doubt that the study has a strong environmental and societal impact. However, when reading the introduction section it is not clear what is the novelty of the paper compared to Belletti et al. (2020) and what is the methodological advance proposed by the authors to assess river connectivity/fragmentation? In the first line of the abstract, the authors mentioned that “it is well established that dams decrease river connectivity”, which is also one of the main results of this study (see Figure 8). Would it be expected that small reservoirs decrease river length fragmentation considering that the high number of small dams will lead to high river fragmentation, as already shown in Belletti et al. (2020)? Despite the different geographical location, dataset, and fragmentation metrics, what is the difference between this study, Belletti et al. (2020), and other studies on the topic?

We agree with the reviewer that we did not do a clear job of explaining the novelty of our work in the original manuscript. We have re-written our introduction to address this point.

Our study has several key differences from Belletti et al. (2020) that we will better highlight in the abstract, introduction, and discussion of the revised manuscript.

1. Belletti et al. (2020) were primarily focused on identifying very small barriers that were not included in existing datasets. Therefore, the analysis they present is primarily on the number of structures they identified and how this impacts the spatial density of structures. In our study, we provide not only the spatial distribution of dams with a more complete dam dataset, but also figure out the temporal evolution of dam constructions over the last century.
2. Belletti et al. (2020) were unable to connect the structures to a river network map because they note that the digital river map for their domain was lacking. They specifically note the need for future work to better evaluate river connectivity. Our work addresses this gap because we map over 50,000 structures on a digital river network map over the whole CONUS. This allows us to evaluate the impact of structures on the river network itself in ways that previous work has not been able to. For example, we are able to calculate river

fragments directly and consider fragment length, presenting a more comprehensive picture on how river connectivity is decreased by dams.

3. Because of our novel approach we can show that mid-sized dams disproportionately contribute to the shortest segments. This finding illustrates that the importance of not just considering large dams.

We would like to acknowledge that there are other works that calculate fragment lengths (e.g. Dynesius & Nilsson, 1994; Grill et al., 2014). Indeed, the reviewer is correct that we follow the methodologies established in those references, as we felt it was best to express our results in terms of established approaches. What sets our study apart from these works is that compared to global studies, ours is the first to include mid-sized dams in this type of analysis. We show that this has a significant impact as mid-sized dams can account for more than 80% of river regulation in some places.

Additional relevant studies include Cooper et al., 2017; Foley et al., 2017; and Poff & Hart, 2002. Cooper et al (2017) and Foley et al. (2017) were primarily focused on other dam metrics and the impacts to fish. Poff & Hart (2002) did a regional scale analysis of small dams but in the context of dam removal. Cooper et al. 2017 is the most similar study to ours, however they only provide metrics for a single point in time. Our work shows the evolution of dam fragmentation over time.

We appreciate the comment from the reviewer and agree that this novelty was not emphasized clearly enough in the original manuscript. As noted above we have significantly revised our introduction and discussion to address this concern.

It would be really interesting to have more information on the main predictors of barrier density (e.g agricultural pressure) and to have analyses and discussions on the environment and hydrological implications of river fragmentation. What is the effect of river fragmentation in a quantitative way (e.g. reduction of flood peak, more frequent dry periods?)?

We agree that this would be interesting additional analysis to conduct. We feel that rigorous analysis on any of the predictors you have noted here would warrant a separate study and is beyond the main purpose of this work. For example, evaluating the impact of structures on flood peaks and streamflows would require a detailed study of reservoir operations. Reservoir operation data is very difficult to come by nationally and is particularly spotty for small structures. In the existing manuscript we do discuss the hydrological implications of fragmentation, including the natural causes of river fragmentation (Lines 160-166) and the need for flood control structures in the wetter regions of the country (Lines 195-197). However, we do agree that some additional discussion and analysis of context would improve the paper. We have added the following discussion of reservoir purpose to the revised manuscript (Lines 264-268 and 278-287):

“Some of this can be explained by reservoir purpose including flood control, hydroelectricity, and water supply. The more humid eastern U. has greater river density and many population centers, so there is more need for flood control. In contrast, the western U.S. is more arid and relies on reservoirs for water supply.”

“Furthermore, since we lack operations data, we use reservoir storage volumes to calculate DOR. We use this metric as an indication of potential river regulation. However, we acknowledge that dam design, reservoir purpose, and passage systems can greatly impact the influence a structure has on a river (Poff & Hart, 2002). For example,

some dams have fish ladders and other structures in place to allow flow of organisms and nutrients. Similarly, the operating strategy of a flood control dam is different than that of a water supply dam. To fully understand the regulatory impact of structures on our streamflow regimes we need a more expansive national structure inventory and accessible dam operations data.”

Results and discussion: I found the results reported in the abstract really vague, without a quantitative indication of the influence of small reservoirs on river fragmentation, and without a clear take-home message from the study. Regarding the discussion and introduction, I would recommend the authors to improve the literature analysis as I had the impression that the authors were biased toward the same references (e.g Belletti et al., Grill et al, and Lehner et al.) in different parts of the paper. In the discussion section, I would include a reference at the end of line 294 to support the statement on the underestimation of regulation and fragmentation on CONUS.

We agree that we were not direct enough in our original manuscript. We have significantly revised our paper in the following ways in response to this concern.

- We rewrote the introduction to be more explicit about the value added by this work.
- We revised the abstract to provide a more concrete summary of our major findings.
- We revised the results and discussion to be more direct and edited many of our subtitles to try to be more specific.
- In response to Reviewer 2, we have also modified our language to be much more explicit about what is and is not included in our study.

Is the first section of the results (the regulation of major rivers in the US) really needed in this study? This result section reads more like a literature analysis and it could be a better fit in the introduction section to provide the reader with the importance of considering dams as drivers for river fragmentation due to their massive presence in the US (as already repeated many times in the paper).

We appreciate the suggestion. We reworked this section to focus on the temporal changes in Degree of Regulation without the commentary of the history of dam construction in the US. This section is now more focused on the figure describing the temporal evolution of river fragmentation over time. We opted to keep this section in the results since it is the first time picturing the fragmentation in CONUS scale over time.

Additional comments:

- Lines 17-29 of the introduction read more like an abstract. Of course, the way to structure the introduction section can be very subjective, but I would first introduce the main problem, then report the previous studies on the topic, then include the methodological gaps, and then describe the novelty and objective(s) of the paper.

We have completely re-written our introduction in response to this comment. Our first paragraph now gets straight to the novelty of our study and then we circle back to previous literature. Also, throughout the introduction we have tried to explain very clearly what has been done in previous research and what value this study adds.

- The current version of the paper has a total of 9 figures. If possible, I would recommend the authors to reduce the number of figures (or merge them) to keep only the most

important and include the others as supplementary. For example, figure 6 is barely referred to in the paper and it could go as supplementary.

We decided to move Figures 4 and 6 to Supplementary Information. We kept Figures 7 and 8 (now 5 and 6) because they are most directly related to main points of the revised manuscript.

- In order to ensure transparency and replicability, data and scripts used in this study have to be reported in the corresponding sections.

All our analysis codes are available in a GitHub repository https://github.com/rspinti/medium_dam_fragmentation_regulation. We have added links to the Code Availability section. Additionally, within our repository we have a readme which clearly designates how to reproduce each figure and where the code is located for each result. Our analysis is based on publicly available datasets. We include links to every dataset used and the version number within our GitHub repo (as well as citations in our methods section). We have also added a note to the Data Availability section of the revised manuscript making this clear.

- Lines 311-320 should be moved in the introduction to define since the beginning of the paper the meaning of river fragmentation also for non-expert readers.

We appreciate the recommendation and have added the definitions to the introduction, so readers understand how we are using these terms in the paper (Lines 40-42 ad 48-50).

- Figure 4. Include the years in the figure and not only in the caption.

In response to later reviewer comments about reducing the number of figures in the paper, we have decided to remove this figure.

- Line 210, 73%, while 70% in line 305. Try to be consistent with the percentages.

We thank the reviewer for finding this inconsistency and have fixed Line 257 to say 73%.

Response to comments by anonymous Reviewer 2

We have addressed their comments (bolded) below.

In my opinion the authors have accomplished a very good study. For a particular dataset they have performed an elegant analysis of changes in flow/storage over time, and the cutting of river segments into more and smaller links, termed river fragmentation. In essence, they have done a very nice job with an existing data base, mapping tools, and by employing some widely used metrics. Lit review, writing, analysis, are all very well done. I follow this literature, and found this manuscript to be a solid contribution.

We would like to thank the reviewer for their comments and for acknowledging the promise of our work. We are excited to contribute to this subset of literature.

My main concern is a sense of over-selling the significance of their study, or perhaps just that they are insufficiently forthcoming about the limitations. The analysis uses an existing data base of dams maintained by the Army Corps and digitized by Ostroff et al (2013). As the authors point out, there are just over 50,000 dams in this data base, but only 1945 of the largest are included in world data bases – hence they are performing the first analysis of “small dams” in the U.S.

We appreciate the comment and agree with the reviewer that while our dataset includes many smaller dams, it is still missing the smallest structures that can be barriers to flow (and in some cases may not even be classified as dams). As detailed in our response below we have modified our language throughout the manuscript to better clarify what we included and critically what we are missing here. Most significantly we have decided that the most transparent way to handle this is to talk about the dams in the national inventory that we are including as ‘mid-sized’ or ‘medium’ dams rather than small dams. We now include a clear definition of exactly the size thresholds for our study and we have added text in both the introduction and discussion to explain the small dams we are missing and why it is currently impossible to include them in this type of work.

The key data set is from Ostroff, A., Wieferich, D., Cooper, A., Infante, D., & Program, U. A. G. (2013). 2013, 2012 National Anthropogenic Barrier Dataset (NABD). In. U.S. Geological Survey - Aquatic GAP Program: Denver, CO. No link is given, and it is not found in google scholar (ie. No citations to it). It can be found at: <https://www.sciencebase.gov/catalog/item/get/512cf142e4b0855fde669828>. It was derived from point dataset of the 2009 National Inventory of Dams (NID) created by the U.S. Army Corps of Engineers (USACE). Language in the description indicates that these are all in fact large (but perhaps not “very large”) dams.

We thank the reviewer for pointing out that the citation for the dataset we used needs to be updated. We have updated the citation for NABD per the comment and the new citation can be found in the References.

In response to the note about the language in the NABD description online, we agree that NABD did not contain all the “very large” dams. We added some of these dams into our dataset from GRanD during our data processing to ensure the largest dams were included. These lines covered in our methods section (Lines 321-323 of the revised manuscript) detail this process.

Given that Poff and Hart 2002 (not cited) estimate something of the order of two million small dams in the U.S. (and other studies point to additional barriers formed by poorly designed culverts affecting headwater streams (not cited)), the authors’ claim that they are assessing small dams, and that this is a something like a complete dataset of small dams, should be far more qualified. For comparison, the Belletti paper they cite analyzes over one million dams in Europe, further indicating the very substantial undercount in this study. Based on the authors’ presentation, readers of this study might reasonably conclude that all or nearly all small dams in the U.S. are included (Line 278 “This study is the first national evaluation which takes all structures into account.”). This is wildly untrue. It is understandable that scholars whose skill set lies in hydrological and geospatial analysis can only perform their work on suitable and available datasets. But they should not mislead their readers into thinking the underlying dataset is more exhaustive than it is. Line 293 “reveals the full extent” Using this language here is not justified.

The lower limit for dams to be included in the National Inventory of Dams (which is the baseline dataset for our work is “equal to or exceed 25 feet in height and exceed 15 acre-feet in storage, or equal to or exceeding 50 acre-feet storage and exceeding 6 feet in height”¹. While we do still think that this is a dramatic improvement over the large dams that are included in previous global studies, we agree with the reviewer that this does not include all dams in the US

¹ <https://nid.usace.army.mil/#/what-is-nid/closer-look>

and it certainly does not include all flow barriers. We agree that our choice of language here was unclear and that we need to do a better job of qualifying our results. In response to this comment, we have:

- Modified our language throughout the manuscript to refer to the dams in the NABD as mid-sized dams and major structures rather than small dams.
- Added paragraphs to both the introduction and discussion clearly explaining the small structures we are missing and the impact this may have.

Clearly the inclusion of an additional 48,000 dams (beyond the roughly 2,000 included in global analyses) will result in greater impacts on storage and fragmentation. However, it is worth noting that river fragmentation as used here lies in the algorithm used to estimate it. This study wishes to argue that number and placement of dams (ie, what they analyze) are more important, than size, dam design and operation, etc. They offer some caveats, then make this claim on Line 273: “Ultimately, dam impacts to ecosystems depend more on number and location of barriers, not their size (Belletti et al., 2020).” I haven’t consulted this source but am skeptical of the claim, which is presented here as received knowledge. Obviously, it supports a view that the authors’ analysis is not undermined by details of dam size, design, and operation. However, they do not make the case for this claim on line 273.

We agree that our wording here was not accurate. The point we wanted to make was that any structure that impedes flow can have an environmental impact by causing fragmentation. The point made by Belletti et al. (2020) was that we often focus on the largest dams because they have the greatest potential to impact flow regimes, but that any small structure on a river network decreased connectivity and can impact sediment transport, movement of aquatic organisms and river communities. Additionally, the cumulative impact of many small structures on river connectivity can be much larger than a small number of very large dams. In response to this comment and other reviewer comments we have completely revised our introduction and discussion to be more clear on these points.

The authors do include brief mention of caveats, but in my view these are inadequate. Size of dam, design and operation, and passage devices vary considerably and are likely to profoundly influence physical (sediment transport), chemical (nutrient and materials transport) and biological (movement of organisms) dam impacts. Lines 280-288 indeed acknowledge some of these concerns, but they have the feel of sweeping under the carpet some major reservations that deserve more direct discussion.

We thank the reviewer for their comment. We agree that more clarification is needed about the caveats of dams. We have added the following two paragraphs to the end of our discussion section and more discussion of limitations in our introduction and in the results section on DOR:

“This study is the first national evaluation which accounts for all nationally inventoried structures that can be spatially linked to the river network. While NABD is more complete than other global dam databases, it is focused on dams with the largest potential to cause harm and has a lower threshold for volume and storage height. Thus, it is not a comprehensive database of all anthropogenic structures. There are likely millions of small and very small structures that further fragment our river systems (Graf,

1993). Thus, our results should be viewed as a conservative estimate of river fragmentation caused by major structures.

Furthermore, since we lack operations, we use reservoir storage volumes to calculate DOR. We use this metric as an indication of potential river regulation. However, we acknowledge that dam design, reservoir purpose, and passage systems can greatly impact the influence a structure has on a river (Poff & Hart, 2002). For example, some dams have fish ladders and other structures in place to allow flow of organisms and nutrients. Similarly, the operating strategy of a flood control dam is different than that of a water supply dam. To fully understand the regulatory impact of structures on our streamflow regimes we need a more expansive national structure inventory and accessible dam operations data.” Revised manuscript lines (269-287)

Another concern (although not terribly serious in my view) is that the findings are not very surprising. Lines 196-200: a major finding is that river fragmentation due to dam construction increased more in humid areas than in arid areas. While nicely demonstrated by their analyses and graphics, this basically says that more dams have been built on rivers draining wetter landscapes - hardly a surprise, as the authors themselves note. Line 211 It also is unsurprising 48,000 ‘small’ dams account for more river fragmentation than 2000 ‘large’ dams – there are more of them.

We acknowledge that it is true that our findings do not contradict what we expected. However, we find the novelty of this study to be that we were able to map spatially and temporally the scope of this problem. Our most significant finding is that when you take into account all major structures human development has actually reversed natural aridity patterns that would drive river fragmentation. We have rephrased some of our results including the following to explain the novelty of our work. We also have revised the abstract per suggestions of our other reviewer to highlight the novelty of our study.

Minor but perhaps worth noting: This is not the first analysis of the NABD database. I have not attempted an exhaustive search, but here I offer one example. Perhaps the authors should note such works. The 50,000 dams are mapped and their distribution discussed here (Foley MM, Magilligan FJ, Torgersen CE, Major JJ, Anderson CW, Connolly PJ, et al. (2017) Landscape context and the biophysical response of rivers to dam removal in the United States. PLoS ONE 12(7): e0180107. <https://doi.org/10.1371/journal.pone.0180107>).

We appreciate the reviewer pointing out additional studies. We reviewed this study and similar studies (Cooper et al., 2017; Perkin et al., 2015) to determine how our use of NABD differs from theirs. All these studies were primarily focused on dam impacts to fish and calculating relevant metrics. Cooper et al. (2017) do calculate Degree of Regulation in their study, but do not have the temporal aspect that we show in our study. We have added a discussion of these studies to our revised introduction.

Nit-picking: Line 141 fig 4 c and 4d are mentioned before 4a and 4b.

We appreciate the reviewer pointing this out to us. In response to an earlier comment, we decided to remove Figure 4 to decrease our number of figures.

REVIEWERS' COMMENTS

Reviewer #1 (Remarks to the Author):

I would like to thank the authors for having carefully addressed all my comments. I believe that the manuscript can be accepted for publication

Reviewer #2 (Remarks to the Author):

The revised manuscript satisfactorily addresses all concerns raised. This study advances our understanding of the spatial distribution and likely environmental consequences of river fragmentation resulting from mid-sized dams in the United States, including an anthropogenic-driven shift to greater fragmentation in humid regions

J D Allan